# Oxidative and Glycolytic Metabolism: Their Reciprocal Regulation and Dysregulation in Cancer

**DOI:** 10.3390/cells14151177

**Published:** 2025-07-30

**Authors:** Marco Cordani, Cristiano Rumio, Giulio Bontempi, Raffaele Strippoli, Fabrizio Marcucci

**Affiliations:** 1Department of Biochemistry and Molecular Biology, Faculty of Biological Sciences, Complutense University of Madrid, 28040 Madrid, Spain; mcordani@ucm.es; 2Instituto de Investigación Sanitaria San Carlos (IdISSC), 28040 Madrid, Spain; 3Department of Pharmacological and Biomolecular Sciences, University of Milan, Via Trentacoste 2, 20134 Milan, Italy; cristiano.rumio@unimi.it; 4Department of Molecular Medicine, Sapienza University of Rome, Viale Regina Elena 324, 00161 Rome, Italy; giulio.bontempi@inmi.it (G.B.); raffaele.strippoli@uniroma1.it (R.S.); 5Gene Expression Laboratory, National Institute for Infectious Diseases, Lazzaro Spallanzani, IRCCS, Via Portuense, 292, 00149 Rome, Italy

**Keywords:** glycolysis, oxidative metabolism, regulation, cancer, lactic acidosis

## Abstract

Oxidative and glycolytic metabolism produce energy in the form of ATP and produce intermediates for biomass production. Oxidative metabolism predominates under normoxic conditions and in quiescent or slowly proliferating cells. On the other hand, under hypoxic or pseudohypoxic conditions and in rapidly proliferating cells, glycolysis becomes the predominant pathway. The balance between oxidative and glycolytic metabolism is finely tuned in physiological conditions and becomes dysregulated in many pathological conditions, most notably cancer. In this article we summarize the evidence that has been gathered over the last few years on the mechanisms underlying this balance and the consequences of their dysregulation. We discuss first the non-metabolic factors (mitochondria, cell cycle, cell type, tissue type), then molecules that are at the intersection between glycolytic and oxidative metabolism and those molecules that are inherent to oxidative or glycolytic metabolism that affect the equilibrium between the two energy-producing pathways. Eventually, we discuss pharmacologic or genetic means that allow manipulating this equilibrium. As will be seen, lactic acidosis has taken center stage in this field and lactate has been shown to fuel oxidative metabolism. This suggests that if glycolytic metabolism predominates, as has often been shown in cancer, mechanisms come into work that reestablish a metabolic heterogeneity. Thus, while one pathway may be predominant over the other, it seems as if fail-safe mechanisms are at work that avoid the possibility that it becomes the only energy-producing pathway. Eventually, we discuss possible therapeutic consequences that may derive from this expanding knowledge, in particular, as regards tumor therapy.

## 1. Introduction

Oxidative metabolism (i.e., the tricarboxylic acid (TCA) cycle and oxidative phosphorylation (OXPHOS)) and glycolysis are the metabolic pathways that produce energy in the form of adenosine triphosphate (ATP) (Figure 1). Oxidative metabolism is fueled mainly by glucose, fatty acids (FAs), glutamine and, as will be discussed in detail later in this article, lactate [1]. Glucose fuels oxidative metabolism after being converted to pyruvate through glycolysis. Pyruvate is then converted by pyruvate dehydrogenase (PDH) into acetyl-CoA, which is the starting point of the TCA cycle that generates CO_2_, nicotinamide adenine dinucleotide, reduced (NADH) and flavin adenine dinucleotide, reduced (FADH). These molecules then enter OXPHOS, which eventually generates ATP [2]. The term glycolysis is used (in fact inappropriately because glycolysis terminates with the production of pyruvate) when under anaerobic conditions pyruvate is fermented to lactate by lactate dehydrogenase (LDH) with the generation of NAD, oxidized (NAD^+^), which is required to promote glycolysis at the level of glyceraldehyde 3-phosphate dehydrogenase (GAPDH). Almost a century ago, Otto Warburg found that in tumor cells, glycolysis can lead to the generation of lactate even under aerobic conditions [3]. This phenomenon is referred to as aerobic glycolysis or the Warburg effect and is now considered one of the hallmarks of cancer [4]. Aerobic glycolysis, however, can also occur in non-transformed cells undergoing rapid proliferation like mitogen-stimulated lymphocytes [5].

In contrast to rapidly proliferating cells, slow-cycling or quiescent cells show a predominantly oxidative metabolism. Accordingly, blocking glycolysis with 3-bromopyruvate was strongly inhibitory on rapidly proliferating cells but less so on slowly proliferating cells [6,7].

Oxidative metabolism is much more efficient than glycolysis in generating ATP (up to 36 ATP molecules vs. 2 ATP molecules). In fact, oxidative metabolism generates more than 90% of the ATP under normoxic conditions [8]. Given that, it is not yet entirely clear why many, but not all, tumor cells or rapidly proliferating cells recur to glycolysis as the predominant energy-producing pathway. It has been argued that tumor cells often resort to glycolysis for energy production because, despite the lower yield, ATP is produced at a faster rate than by oxidative metabolism [9,10]. Glycolysis may also give rise to increased levels of metabolites that feed side metabolic pathways that are crucial for tumor growth such as the pentose phosphate pathway (PPP), which leads to the generation of nucleotides and antioxidants [11]. Moreover, by reducing the impact of OXPHOS, glycolysis upregulation may protect tumor cells from oxidative stress, since OXPHOS is a major source of reactive oxygen species (ROS). It has also been shown that aerobic glycolysis can be driven when the demand for NAD^+^ to support oxidation reactions exceeds the rate of ATP turnover [12]. In this case, NAD^+^ becomes limiting, thereby favoring the fermentation of pyruvate to lactate even when oxygen is available. In fact, in models of primary solid tumors, oxidative metabolism was found to be downregulated, while glycolysis was increased [9]. This increase, however, did not compensate for the decreased oxidative metabolism in terms of ATP production. These results suggest that in these tumor models, glycolysis was upregulated mainly to discharge ATP-independent functions like the generation of intermediates for biomass production.

It should be considered, however, that the predominance of aerobic glycolysis in tumors is not as overarching as originally thought. In fact, many tumor cells of different tumor types show a predominantly oxidative metabolism. Thus, a study performed on 31 cancer cell lines or tissues showed that the average ATP contribution from glycolysis was 17% [13]. Another study showed that patient-derived organoids from metastatic pancreatic ductal adenocarcinomas (PDACs) showed subtype-specific differences in glycolysis and oxidative metabolism [14]. We have recently reviewed this topic [1].

Given the putative upsides of glycolysis for tumor growth, it is not clear why, in these cases, oxidative metabolism is the preferred energy-producing pathway. One explanation, which is certainly not exhaustive, is that quiescent or slowly proliferating tumor cells preferentially use oxidative metabolism [15,16], an observation consistent with findings obtained with non-transformed cells [17]. This would also explain the predominance of oxidative metabolism that has been reported for cancer stem cells (CSCs) or metastasizing cells where populations of quiescent/slowly proliferating cells coexist with rapidly dividing cells [18].

It is important to note, however, that while one energy-producing pathway may predominate, oxidative metabolism and glycolysis always coexist in the same tumor cell population [19]. In fact, it has been argued that a hybrid, oxidative/glycolytic state may allow tumor cells to better adapt to different conditions existing in the tumor microenvironment (TME) [20] and promote their metastatic dissemination and resistance to insults, including drug resistance [21]. Metabolic heterogeneity can exist within and between tumor cells and the same tumor cells can switch from a predominantly oxidative metabolism to a predominantly glycolytic metabolism [22]. The switch from one energy-producing pathway to the other is dictated by a large number of tumor cell-intrinsic or -extrinsic stimuli, including hypoxia and glucose shortage [23], that have been recently reviewed [1,24]. These stimuli induce quantitative (e.g., overexpression) or qualitative changes (e.g., posttranslational modifications) in metabolic enzymes [19,25]. Metabolic reprogramming from oxidative to glycolytic metabolism has been shown not to occur in an abrupt but, rather, stepwise fashion upon transformation of mouse embryonic fibroblasts with H-Ras^V12^/E1A [26]. Initially, the transformation led to a dramatic increase in oxygen consumption and superoxide production and diffuse cell death. During prolonged in vitro culture, glycolytic metabolism gradually increased, while oxidative metabolism decreased. In parallel, cell proliferation increased as well as tumor-initiating potential both in vitro as well as in vivo.

It should be noted that the switch from one energy-producing pathway to the other can be deliberately induced, either by pharmacological means or genetic manipulation. Thus, using PDAC cell lines, it has been shown that the suppression of glycolysis led to the upregulation of oxidative metabolism [27] with glutamine and glutamate being consumed in the TCA cycle. Similarly, Aurora kinase A inhibition downregulated glycolysis that was mediated by the inhibition of MYC targets and activation of peroxisome proliferator-activated receptor alpha (PPAR-α) signaling [28]. Concomitantly, the oxygen consumption rate (OCR) increased due to enhanced fatty acid oxidation (FAO) and overall increased oxidative metabolism. Upregulation of one pathway is accompanied by increased expression of one or more enzymes of the pathway. Overexpression of an enzyme may facilitate its localization to unusual subcellular locations, such as the nucleus, and this location may endow it with non-canonical functions that are referred to as “moonlighting” functions [29,30].

The considerations outlined so far suggest that there must exist a finely tuned balance between oxidative metabolism and glycolysis with several control points that prevent or, if necessary, promote switches from oxidative to glycolytic metabolism or vice versa, depending on the existing physiological or pathological conditions that may occur. In this article, we summarize the novel, expanding knowledge about the factors that control this equilibrium and the dysregulation that may take place under pathological conditions, in particular, in cancer. In order to keep this review in a manageable size, we will not discuss long-standing knowledge about, for example, the allosteric regulation of some glycolytic enzymes (e.g., phosphofructokinase 1 (PFK1), hexokinase (HK)) unless recent insights have expanded our knowledge on these issues. For more information on these aspects, the reader is referred to textbooks or review articles that have been published over the years. 

## 2. Non-Metabolic Factors That Regulate the Equilibrium Between Oxidative and Glycolytic Metabolism

### 2.1. Quantitative Changes in Mitochondria or Components Thereof

Morphologic or quantitative changes in mitochondria have been shown to shift the balance from oxidative metabolism towards glycolysis. Thus, partial depletion of mitochondrial (mt) DNA in SW480 colorectal carcinoma (CRC) cells caused increased glucose uptake and lactate generation [31]. The activity of the glycolytic enzymes HK and PFK1 was upregulated and the cells became resistant to 5-fluoruracil (5FU) and oxaliplatin. When the normal mtDNA content was restored, the cells returned to a predominantly oxidative metabolism and resistance was reversed.

Tumor cells treated with antitumor drugs (paclitaxel or TH287) were arrested in the M phase of the cell cycle, accumulated mitochondria and produced increasing amounts of mitochondrial superoxide [32] that caused oxidative DNA damage. Mitochondrial mass was greatly increased and mitochondrial function inhibited. This induced metabolic reprogramming towards glycolysis.

Mitochondrial fission factor (MFF) is upregulated in hepatocellular carcinoma (HCC) CSCs where it promotes mitochondrial fission and increases tumor-initiating potential. Experimentally induced overexpression of MFF [33] induced mitochondrial fission and metabolic reprogramming from oxidative metabolism towards glycolysis. This shift decreased ROS production, which prevented ROS-mediated degradation of the stem cell transcription factor octamer-binding transcription factor 4 (OCT4). A shift from oxidative to glycolytic metabolism was observed upon knockdown of p32 (also known as p32 gC1q receptor or hyaluronic acid binding protein 1), a protein primarily localized in the mitochondrial matrix [34]. Cells where p32 had been knocked down showed reduced synthesis of mtDNA encoding OXPHOS polypeptides and became poorly tumorigenic in vivo. Expression of exogenous p32 in p32-negative cells restored the original phenotype and tumorigenicity. Thus, in this case and in contrast to other reports, tumorigenicity appears to be linked to the expression of polypeptides involved in oxidative metabolism.

MtDNA is one of the most frequently mutated regions of the cancer genome. Truncating mutations of the mtDNA-encoded complex I gene, Mt-Nd5, were engineered into several mouse melanoma models [35]. The mutations induced the upregulation of glycolysis and a Warburg-like phenotype. Interestingly, tumors bearing mtDNA mutations were sensitized to immune checkpoint inhibitors (ICIs) and showed an improved response rate to ICI therapy compared to mtDNA wild-type tumors.

Extracellular glutamine is an important energy source for many tumors. Glutamine deficiency can occur due to increased metabolic activity in tumors. Among others, it was found to induce the stress-responsive transcription factor DNA damage-induced transcript 3 (DDIT3) [36]. DDIT3 upregulated glycolytic metabolism and, upon translocation to mitochondria, it suppressed OXPHOS through the downregulation of COQ9, a protein necessary for the biosynthesis of coenzyme Q10.

### 2.2. Role of the Cell Cycle

The cell cycle has emerged as an important factor in dictating the choice between a predominantly glycolytic or oxidative metabolism. An early study in yeast showed that genes with metabolic function were differentially expressed depending on the different phases of the cell cycle, with glycolysis predominating during division in order to avoid ROS damage to newly duplicated DNA [37]. A later study showed that mammalian cells also predominantly use oxidative metabolism during the G1 phase of the cell cycle, and glycolysis during the S phase [38]. The choice appeared to be dictated mainly by the timely destruction of isocitrate dehydrogenase (IDH), which shifted the balance towards glycolysis. IDH1 destruction was increased in prostate cancer cells and this favored glycolysis and tumorigenesis.

### 2.3. Cell Type-Dependence

The cell type is another variable that influences the equilibrium between glycolytic and oxidative metabolism. A study investigated the effect of glutamine antagonists in tumor cells of different tumor types (CRC, lymphoma, melanoma) and T lymphocytes [39]. Glutamine carbons enter the TCA cycle upon the conversion of glutamine to glutamate and glutamate to α-ketoglutarate (αKG). The study showed that the inhibition of glutamine metabolism in tumor cells suppressed both oxidative as well as glycolytic metabolism. On the other hand, in effector T lymphocytes, it caused the upregulation of oxidative metabolism and the cells acquired a long-lived, highly activated phenotype. This difference was due to the fact that activated CD8^+^ T lymphocytes, in contrast to tumor cells, upregulated acetate metabolism and generated high levels of acetyl-CoA to fuel the TCA cycle in response to glutamine inhibition. Thus, T lymphocytes were metabolically more flexible in response to glutamine inhibition than the tumor cells investigated in this study.

### 2.4. Tissue Type-Dependence

Predominance of one energy-generating pathway over the other may also depend on the tissue type. Thus, a predominantly glycolytic phenotype in tumors has been shown to depend, at least in part, on the tissue of origin of the tumor [40]. In accordance, lineage-specific transcription factors have been shown to be involved in dictating which type of tumor metabolism predominates [41,42].

## 3. Molecules That Connect or Are Involved in Both Glycolytic and Oxidative Metabolism

In this section we discuss molecules that link glycolytic with oxidative metabolism or that are involved in both metabolic pathways (Table 1).

### 3.1. Pyruvate Dehydrogenase (PDH)

Pyruvate dehydrogenase (PDH) is a gatekeeper enzyme that is part of the PDH complex (PDC) [43]. PDH is the E1α subunit of the PDC and it catalyzes the irreversible decarboxylation of pyruvate. A subsequent step, catalyzed by the PDC E2 subunit, generates acetyl-CoA. As will be seen in the following, the activity of PDH appears to be regulated, among others, by the organization of the mitochondrial respiratory chain. PDH inhibition upregulates glycolytic metabolism. Oncogenic Src, for example, inhibited PDH activity through the direct phosphorylation of Tyr^289^ of the catalytic PDH E1α subunit [44]. This increased glycolysis and lactate production and inhibited ROS production. Expression of a Tyr^289^ non-phosphorylatable PDH E1α mutant in cancer cells with hyperactivated Src restored PDH activity, increased mitochondrial respiration and decreased metastasis formation.

### 3.2. Pyruvate Dehydrogenase Kinase (PDK)

PDK phosphorylates and inactivates the PDH E1α subunit of the PDC [45]. As a result, PDK enhances the conversion of pyruvate to lactate. PDK has four isoforms (PDK1–4). Reduced expression of PDK4 has been shown to divert glucose metabolism towards the TCA cycle during epithelial–mesenchymal transition (EMT) in tumor cells [46]. Accordingly, overexpression of PDK4 partially blocked transforming growth factor (TGF) β-induced EMT and, vice versa, PDK4 inhibition was sufficient to induce EMT and drug (erlotinib) resistance in lung cancer cells. Interestingly, this resistance appeared to be mediated by the interaction between PDK4 and apoptosis-inducing factor, an inner mitochondrial protein. Analysis of human tumor samples showed that PDK4 expression was downregulated in most tumor types.

PDK, in addition to promoting glycolysis through the inhibition of PDH, has also been reported to directly inhibit respiration in tumor cells. Thus, BRAF^V600E^ CRC cells displayed a more fragmented mitochondrial state compared to cells with wild-type BRAF [47]. This effect was induced by mitochondrial PDK1 and caused the upregulation of glycolysis and accelerated tumor growth.

Mitochondrial topoisomerase I is a mitochondrial DNA topoisomerase. Its deficiency has been shown to upregulate PDK4, which is located in the mitochondrial matrix, and the A isoform of LDH (LDHA), which is located in the cytoplasm, thereby leading to glycolysis upregulation, increased tumor cell migration, invasion and metastasis [48]. It appears likely that the upregulation of PDK4 and LDHA is a compensatory consequence of the downregulation of oxidative metabolism that occurs upon the inhibition of mitochondrial topoisomerase I. PDK upregulation has also been shown to occur in response to hypoxia, EGFR activation and the expression of some oncoproteins (K-Ras^G12V^, B-Raf^V600E^) [49]. These conditions were found to induce the mitochondrial translocation of upregulated phosphoglycerate kinase 1 (PGK1), a glycolytic enzyme that catalyzes the dephosphorylation of 1,3-phosphoglycerate to 3-phosphoglycerate with the generation of one ATP molecule. Mitochondrial PGK1 phosphorylated and activated PDK, leading to increased lactate production, reduced ROS production and increased tumorigenesis.

Overexpression of PDK4 and the ensuing upregulation of glycolysis have been investigated in patients with urothelial carcinoma [50]. High expression of PDK4 was associated with negative prognostic factors such as the presence of lymph node metastasis, high tumor grade, vascular invasion, poor disease-free and metastasis-free survival and association with DNA replication and repair.

Mitochondrial translocation of PGK1 in liver CSCs promoted a switch from oxidative to glycolytic metabolism through a PGK1-PDK1-PDH axis [51]. The increased intracellular levels of lactate, in turn, induced the activation of β-catenin/Wnt and promoted self-renewal of liver CSCs. The translocation of PGK1 was induced by a mitochondria-encoded circular RNA, termed mitochondrial circRNA for translocating phosphoglycerate kinase 1, that was highly expressed in liver CSCs. Thus, although mitochondrial localization of PGK1 was the initial inducer of the metabolic switch, also in this case, the final effector mechanism lied in the activation of PDK1 and inhibition of PDH.

### 3.3. Pyruvate Carboxylase (PC)

PC catalyzes the ATP-dependent carboxylation of pyruvate to oxalacetate, which enters the TCA cycle and combines with acetyl-CoA to generate citrate in a reaction catalyzed by citrate synthase. PC is activated by acetyl-CoA, which signals the presence of low levels of oxalacetate. The mechanism underlying the activation of PC by acetyl-CoA has been elucidated [52]. Acetyl-CoA stabilizes PC in a catalytically competent conformation, which promotes ATP hydrolysis and long-distance communication between the two reactive centers. PC contributes to the regulation of gluconeogenesis in the liver, to the synthesis of FAs in adipocytes, to insulin secretion and affords protection from oxidative stress [53]. PC has been shown to be upregulated in a variety of tumors where it supports tumor cell proliferation and tumor progression [54,55].

### 3.4. Oxidized and Reduced Nicotinamide Adenine Dinucleotide (NAD^+^ and NADH)

NAD^+^ is required for the oxidation of glyceraldehyde 3-phosphate to 1,3-biphosphoglycerate catalyzed by GAPDH. On the other hand, NADH is required by OXPHOS for ATP production. It is, therefore, not surprising that enzymes and shuttles that regulate the generation of NAD^+^ and promote the mitochondrial translocation of NADH regulate the balance between glycolytic and oxidative metabolism. Thus, the cytosolic enzymes malate dehydrogenase 1 and glycerol 3-phosphate dehydrogenase 1, which are components of the malate–aspartate shuttle, and the glycerol 3-phosphate shuttle oxidize NADH to NAD^+^ in the cytosol and the shuttles translocate reducing equivalents generated in the cytosol during glycolysis across the inner mitochondrial membrane to be used by OXPHOS [56]. Using NCI-60 cell lines, it has been shown that proliferating cells primarily transform glucose to lactate when glycolysis exceeds the capacity of the mitochondrial NADH shuttles. Increasing their capacity to translocate reducing equivalents decreased glucose fermentation and the generation of lactate and NAD^+^, but did not reduce proliferation [56].

### 3.5. ATP

ATP, whether generated from glycolysis or oxidative metabolism, inhibits PFK1 through an allosteric mechanism [57]. A recent study [58] has shown that the affinity of ATP for the catalytic site of PFK1 is much higher than that for the inhibitory site, but the inhibited ATP-PFK1-ATP complex is much slower in generating the product (i.e., fructose 1,6-biphosphate (F1,6P)) than the uninhibited PFK1-ATP complex. This causes the overall reaction velocity to decrease when the ATP concentration increases.

**Table 1 cells-14-01177-t001:** Molecules that connect glycolytic with oxidative metabolism or are involved in both pathways.

Molecule	Consequences of Its Dysregulation (Overexpression or Inhibition)	References
PDH	PDH inhibition upregulates glycolysis: Src inhibits PDH, upregulates glycolysis, lactate production, inhibits ROS production	[44]
PDK	Inactivation of PDH upregulates glycolysis	[46]
	Inhibits directly mitochondrial respiration	[47]
	Upregulated in response to mitochondrial topoisomerase I deficiency, hypoxia, EGFR activation, oncoproteins	[48,49]
	High expression associated with negative prognosis in patients	[50]
PC	Upregulated in tumors where it supports tumor cell proliferation and tumor progression	[54,55]
NAD^+^/NADH	Cells resort to aerobic glycolysis when capacity of NADH shuttles is exceeded	[56]

Abbreviations: ATP, adenosine triphosphate; EGFR, epidermal growth factor receptor; NAD^+^, nicotinamide adenine dinucleotide, oxidized; NADH, nicotinamide adenine dinucleotide, reduced; OXPHOS, oxidative phosphorylation; PC, pyruvate carboxylase; PDH, pyruvate dehydrogenase; PDK, pyruvate dehydrogenase kinase; ROS, reactive oxygen species; TCA, tricarboxylic acid.

## 4. Factors Inherent to Oxidative Metabolism That Control the Equilibrium with Glycolysis

### 4.1. Different Forms of Mitochondrial Respiratory Complexes (MRCs)

It has been recently shown that two separate MRC organizations that are differently responsive to glycolysis coexist in human cells and postmitotic tissues [59] (Table 2). The two MRC forms were dubbed C-MRC and S-MRC and depend on the preferential expression of three COX7A subunit isoforms, COX7A1/2 and SCAFI (COX7A2L). The prevalence of each MRC organization appeared to be regulated by the activation state of the PDC. Under oxidative conditions, the C-MRC was more efficient, whereas the S-MRC maintained OXPHOS once reprogramming towards glycolysis had occurred. This is an important observation since it demonstrates the existence of mechanisms that prevent a complete suppression of oxidative metabolism when a switch towards glycolysis occurs, thereby promoting cell-intrinsic metabolic heterogeneity.

### 4.2. α-KG

α-KG can act as an electron donor for the oxygen sensors prolyl hydroxylases (PHD) [60]. Under normoxic conditions, PHDs hydroxylate HIF1α, thereby leading to its degradation through the von Hippel–Lindau-mediated ubiquitin–proteasome pathway. Under hypoxic or pseudohypoxic conditions, HIF1α is stabilized and, after binding to the HIF1β subunit, upregulates the expression of several genes involved in glycolytic metabolism. The introduction of α-KG derivatives has been shown to restore normal PHD activity, thereby preventing HIF1α-mediated upregulation of glycolysis [60].

### 4.3. ROS

The majority of ROS are side products of OXPHOS. It has been shown that ROS induce EMT and a glycolytic switch in MCF-7 breast cancer cells [61]. In parallel, ROS inhibited OXPHOS as well as cytochrome c oxidase inhibition. These effects were due to the activation of the homeobox transcription factor distal-less homeobox-2, which, in turn, activated the EMT master transcription factor Snail. Similarly, ROS induced the upregulation of glycolytic metabolism in acute myeloid leukemia (AML) cells [62]. This was the result of ROS-induced expression of uncoupling protein 2 (UCP2) and phosphorylation of AMP-activated protein kinase (AMPK). These effects, in turn, upregulated the expression of the PFKFB3 isoform of PFK2. Overexpression of PFKFB3 upregulated glycolysis and increased cell proliferation.

### 4.4. IDH

IDH is an enzyme of the TCA cycle that catalyzes the formation of oxalosuccinate from isocitrate and α-KG from oxalosuccinate. Increased IDH1 activity, due to enhanced homodimerization of the enzyme driven by a long non-coding (lnc) RNA, led to increased production of αKG and decreased generation of ROS [63]. These changes were paralleled by a compensatory downregulation of glycolysis. C-MYC repressed the expression of the lncRNA and this promoted glycolysis upregulation.

*IDH1/2* are mutated in most lower-grade gliomas and 10–20% of AMLs. Mutant IDH1/2 lose the activity of wild-type IDH1/2 and, instead, convert α-KG to the R enantiomer of 2-hydroxyglutarate (2HG) instead of oxalosuccinate. The consequences of *IDH1/2* mutation on glycolytic metabolism are contradictory, with some evidence showing suppressive effects, while other evidence suggests stimulatory effects on glycolysis. Thus, 2HG has been shown to inhibit the post-transcriptional upregulation of the platelet isoform of PFK1 (PFKP) and the B isoform of LDHB (LDHB) in leukemia cells, thereby suppressing aerobic glycolysis [64]. This effect was not observed in CD34^+^ hematopoietic stem/progenitor cells. In *IDH* mutant glioblastoma multiforme (GBM) cells, HIF1α-responsive genes, including those encoding glycolytic enzymes, were underexpressed [65]. *LDHA* was silenced in IDH mutant GBM CSCs and silencing was accompanied by increased methylation of the *LDHA* promoter. These results suggest that IDH mutant GBMs have limited glycolytic capacity and may be at the origin of their slow growth and better prognosis.

Other studies showed that IDH1/2 mutations lead to glycolysis upregulation. Thus, mutant IDH1 was found to upregulate the expression of PFKP through aberrant histone modification, and this led to the overall upregulation of glycolysis [66]. Another study showed that a gain-of-function IDH2 mutation promoted reductive glutamine metabolism that led to the stabilization of HIF1α [67] in gemcitabine-resistant urothelial carcinoma cells. This, in turn, stimulated glycolysis and caused cross-resistance to cisplatin by increasing antioxidant defense via increased nicotinamide dinucleotide phosphate, reduced (NADPH) and glutathione production. Inhibition of IDH2 restored chemosensitivity. 

The reasons as to why *IDH* mutations may lead to glycolysis downregulation or upregulations are unclear. It may be speculated that the levels of 2HG that are produced as a result of these mutations may vary depending on temporal and/or geographical issues and these different levels may induce predominantly upregulated or downregulated glycolysis. As we will see in the following, similar effects have also been described for metformin.

### 4.5. Succinate Dehydrogenase (SDH)

Upregulation of glycolysis occurs upon the downregulation or mutation of individual enzymes involved in oxidative metabolism. This has been shown for SDH, a mitochondrial enzyme involved in both the TCA cycle and OXPHOS [68]. Thus, knocking out the *Sdhb* subunit in mouse ovarian cancer cells upregulated glycolysis with cells taking up more glucose and increasing lactate generation. This was accompanied by the upregulation of genes promoting hypermethylation of the epigenome, genes involved in PPP and nucleotide metabolism, while the expression of enzymes involved in OXPHOS was downregulated. The cells showed enhanced proliferation and EMT.

### 4.6. Fumarate Hydratase (FH)

FH catalyzes the reversible hydration of fumarate to malate in the TCA cycle. Hereditary leiomyomatosis and renal cell cancer are due to biallelic inactivation of the gene encoding FH. This leads to fumarate accumulation, which inactivates factors that are involved in the replication of mtDNA. This causes the loss of respiratory chain components, and, consequently, a shift to aerobic glycolysis and diversion of glucose to the PPP [69].

### 4.7. ATP Citrate Lyase (ACLY)

ACLY converts cytosolic citrate to acetyl-CoA and oxalacetate and is often upregulated in tumors [70]. ACLY activity decreases citrate levels. This leads to a reduction in the inhibitory effect of citrate on PFK1 and PFK2 [71] and to increased levels of F1,6P. This promotes the upregulation of glycolysis and downregulation of oxidative metabolism [72]. Interestingly, ACLY can be directly activated by phosphorylated fructose [73].

### 4.8. Citrate Synthase and Citrate

Citrate synthase catalyzes the conversion of oxalacetate and acetyl-CoA to citrate. Citrate synthase inhibition in tumor cells upregulates aerobic glycolysis and induces EMT [74]. On the other hand, high cytosolic concentrations of citrate inhibit glycolysis [75]. Citrate has been found to inhibit the proliferation and induce the apoptosis of various solid tumor cells at high concentrations [76] and to exert antitumor activity in vivo either alone or in combination with chemotherapeutics [77]. Whether these effects are the consequence of glycolysis inhibition due to the allosteric inhibitory effect of citrate on PFKs is unclear.

**Table 2 cells-14-01177-t002:** Molecules of oxidative metabolism that regulate the equilibrium with glycolysis.

Molecule or Molecular Complex	Consequences of Its Dysregulation (Overexpression or Inhibition)	References
Different forms of mitochondrial respiratory complexes (MRCs)	C-MRC more efficient under oxidative conditions S-MRC maintains OXPHOS upon reprogramming towards glycolysis	[50]
α-KG	α-KG can act as an electron donor for PHDs. It restored normal PHD activity and prevents HIF1α-induced glycolysis upregulation	[60]
ROS	ROS induced upregulation of glycolysis in breast cancer cells and in AML cells	[62]
IDH	Increased IDH1 activity led to increased production of αKG and decreased generation of ROS	[63]
	Mutated IDH1/2 have been found to either upregulate or downregulate glycolysis	[64,65,66,67]
SDH	Downregulation of SDH in ovarian cancer cells upregulated glycolysis, lactate production, enhanced proliferation and induced EMT	[68]
FH	Downregulation of FH causes loss of respiratory chain components and a shift to aerobic glycolysis	[69]
ACLY	Increased ACLY decreased citrate levels, upregulated glycolysis and downregulated oxidative metabolism	[70,71,72,73]
Citrate synthase and citrate	Citrate synthase inhibition in tumors upregulated glycolysis and induced EMT	[74]
	High cytosolic citrate concentrations inhibited glycolysis	[75]
	Exogenous addition of citrate had antitumor effects	[76,77]

Abbreviations: ACLY, ATO citrate lyase; αKG, α-ketoglutarate; AML, acute myeloid leukemia; EMT, epithelial–mesenchymal transition; FH, fumarate hydratase; HIF, hypoxia-inducible factor; MRC, mitochondrial respiratory complex; OXPHOS, oxidative phosphorylation; PHD, prolyl hydroxylase; ROS, reactive oxygen species.

## 5. Factors Inherent to Glycolytic Metabolism That Control the Equilibrium with Oxidative Metabolism

### 5.1. HK

HK catalyzes the first step of glycolysis that leads to the formation of glucose 6-phosphate (G6P) from glucose and ATP [78]. The HK1 isoform is localized on the outer mitochondrial membrane in juxtaposition with voltage-dependent anion channel 1. The ATP that is extruded by these channels is utilized by HK for the generation of G6P [79]. It was recently reported that HK1 forms rings around mitochondria that prevent mitochondrial fission [80] (Table 3). Formation of these rings is promoted by low levels of ATP and G6P. Prevention of mitochondrial fission preserves the integrity of mitochondria under conditions of energy stress. Accordingly, the presence of these HK1 rings appeared to rewire cellular metabolism towards increased TCA cycle activity. The activity of HK1 can also be modulated by O-linked β-N-acetylglucosaminylation (O-GlcNAcylation) [81]. O-GlcNAcylation affects the regulatory domain of HK1 and promotes the localization of HK1 on the outer mitochondrial membrane, leading to increased glycolytic as well as mitochondrial ATP production. Interestingly, the HK2 isoform did not undergo a similar posttranslational modification, most likely because HK2 lacks the regulatory domain of HK1.

On the other hand, the inhibition of HK2 causes the inhibition of glycolysis and a compensatory upregulation of oxidative metabolism. Thus, the dissociation of HK2 from the outer mitochondrial membrane inhibited its enzymatic activity and glycolysis and promoted the upregulation of oxidative metabolism [82]. Dissociation was caused by mechanistic target or rapamycin (mTOR), which relocated to mitochondria in response to a single-dose irradiation. The relocation promoted the formation of a complex of mTOR with HK2, which caused the dissociation of HK2. Normal hepatocytes express glucokinase as the predominant form of HK. HCC cells suppress glucokinase expression, but upregulate HK2 expression [83]. Deletion of HK2 in these cells inhibited tumor incidence in a mouse model of hepatocarcinogenesis. In human HCC cells, the silencing of HK2 inhibited tumorigenesis, increased cell death, downregulated glycolysis and caused a parallel enhancement in oxidative metabolism.

### 5.2. PFK1

PFK1 converts fructose 6-phosphate (F6P) to F1,6P in the first committed step of glycolysis. PFK1 is activated by signals that reflect a low-energy state in the cell such as AMP and ADP, and, on the other hand, it is inhibited by signals that reflect a high-energy state such as ATP and citrate [57]. Oligomerization regulates PFK1 activity with activators inducing the tetrameric, active form of the enzyme and inhibitors promoting the disassembly of the tetramers. PFK1 exists in three isoforms: a platelet isoform (PFKP), a liver isoform (PFKL) and a muscle isoform. PFKP is the predominant tumor-associated isoform [84]. It is upregulated in many tumor types where it promotes aerobic glycolysis, proliferation and metastasis formation [85].

### 5.3. Phosphoglycerate Mutase (PGM)

PGM catalyzes the conversion of 3-phosphoglycerate to 2-phosphoglycerate in the glycolytic pathway. The effects of the persistent overexpression of PGM were investigated on cardiac energy and metabolism in mice [86]. Increased levels of some glycolytic metabolites upstream and downstream of PGM2 were observed. On the other hand, the levels of metabolites generated during the initial steps of glycolysis and of lactate remained unchanged. Inhibition of respiration in mitochondria and increased generation of ROS were also observed. These changes were accompanied by reduced stress resistance in mice. These effects are also of interest in the cancer setting since PGM is often overexpressed in tumors [87].

### 5.4. Pyruvate Kinase M (PKM)

PK catalyzes the final step in glycolysis, with the transfer of a phosphate group from phosphoenolpyruvate to ADP, with the production of pyruvate and ATP. Mammals express four PK isozymes: PKM1, PKM2 (the most prevalent in cancer cells), L and R. As to the two M isoforms, PKM1 is a constitutively active variant that is often expressed in differentiated tissues, while PKM2 exists as a low-activity dimer that converts to an active tetramer upon binding of F1,6P. PKM2 is mainly expressed in rapidly proliferating cells, like tumor cells.

The stable knockdown of PKM1 and PKM2 isoforms was investigated in H1299 and A549 lung carcinoma cell lines [88]. The knockdown of both isoforms in A549 cells significantly reduced the cellular ATP level, while it was unaltered in H1299 cells. The reason for this differential response was that PKM knockdown in H1299 cells led to AMPK activation and, consequently, to autophagy induction and the stimulation of mitochondrial biogenesis. These effects allowed maintaining energy homeostasis. In contrast, knocking down either of the PKM isoforms in A549 cells, which lack liver kinase B1, an AMPK activator, failed to activate AMPK and induced apoptosis. Thus, the metabolic balance in H1299 cells upon knockdown of PKM was maintained through AMPK-mediated metabolic reprogramming towards oxidative metabolism. A recent study reported results that are, at least in part, consistent with those just discussed [89]. PKM2 was found to bind to the histone methyltransferase enhancer of zeste homolog (EZH) 2 in the nucleus of triple-negative breast cancer (TNBC) cells. The complex localized to the nucleus where it repressed the expression of a transporter that promotes the mitochondrial import of long-chain FAs and, consequently, FA oxidation (FAO). Inhibition of the PKM2-EZH complex induced reprogramming from glycolysis towards FAO. Altogether, these results suggest that PKM uses different mechanisms to upregulate/preserve glycolytic metabolism.

Expression of active PKM2 has been shown to reduce lactate production in spite of it being overexpressed in tumor cells. This paradoxical effect has now been clarified [90] by showing that oxalacetate synthesis is increased upon PKM2-induced activation of glutamate pyruvate transaminase 2, which converts pyruvate and glutamate to alanine and αKG. This causes increased α-KG flux through the TCA cycle and, consequently, to increased levels of oxalacetate. Increased levels of oxalacetate, in turn, competitively inhibit LDHA in tumor cells.

### 5.5. Fructose 1,6-Biphsophate (F1,6P)

In many cells, oxidative metabolism is inhibited by high concentrations of glucose, a phenomenon referred to as the “Crabtree effect” [91]. F1,6P, a glycolytic metabolite, was found to mediate this effect in yeast and isolated liver rat mitochondria [92]. Thus, physiological concentrations of glucose 6-phosphate and fructose 6-phosphate stimulated respiration and this effect was antagonized by F1,6P. The results also suggested that F1,6P might have a direct regulatory role on the mitochondrial respiratory chain since it inhibited electron transfer through electron transport chain (ETC) complexes III and IV. This caused a reduction in ATP and citrate production and, consequently, reduced inhibition of PFK2 and glycolysis.

### 5.6. Lactate and Lactic Acidosis

It is somehow surprising to note that the final product of fermentative glycolysis, lactic acid, i.e., lactate in combination with H^+^ protons, is a main driver of oxidative metabolism in tumor cells. This represents an excellent example of how tumors apparently aim to achieve metabolic heterogeneity and avoid an excessive predominance of one energy-producing pathway over the other. In fact, it appears that once tumor cells are driven to a predominantly glycolytic metabolism, this same metabolism lays the foundation to reestablish a more balanced equilibrium with oxidative metabolism.

Lactate is extruded into the microenvironment (TME in case of tumors) in association with H^+^ protons as lactic acid when fermentative glycolysis predominates and the intracellular pH is driven towards acidity. This leads to the generation of an acidic extracellular microenvironment [93]. Mitochondrial production of CO_2_ and its dissociation into HCO_3_^−^ and H^+^ can also contribute to the acidification of the extracellular microenvironment [94]. Lactic acidosis occurs when lactate concentrations exceed 15 nM and the pH of the TME is in the range of 5.8–6.7. Hyperlactemia in the absence of acidosis does not occur in vivo, but can be created in vitro by adding sodium lactate to the culture medium [95].

Circulating lactate has long been considered a waste product of fermentative glycolysis. Instead, it is now looked at as a pivotal carbon source for the TCA cycle. Thus, the contribution of circulating lactate to TCA cycle intermediates in tumors exceeded that of glucose, with glutamine exceeding the contribution of lactate only in pancreatic tumors [96]. Lactate was also shown to be a carbon source for the TCA cycle in non-small-cell lung cancer [97]. In mice, the depletion of monocarboxylate transporter 1 (MCT1) from tumor cells inhibited the contribution of lactate as fuel for oxidative metabolism, showing that lactate transport by tumor cells was required for this purpose. Importantly, the contribution of lactate to the TCA cycle in this setting predominated over that of glucose. Acute myeloid leukemia (AML) patients with FMS-like tyrosine kinase 3 (FLT3) internal tandem duplications cells showed high activity of ETC complex II and correspondingly high mitochondrial respiratory activity [98]. In response to the inhibition of ETC complex II, AML cells imported lactate from the TME and used it for mitochondrial respiration.

Lactate-induced metabolic reprogramming can also be the consequence of endocrine signaling. Insulin has been shown to upregulate glycolysis and increase lactate production [99]. Insulin also promoted lactate oxidation in the TCA cycle by reducing the levels of circulating FAs, which competed with lactate for mitochondrial oxidation. High levels of lactate, however, inhibited insulin-induced glycolysis and decreased lactate production by lowering circulating FAs via the G protein-coupled receptor 81 (GRP81)/hydroxycarboxylic acid receptor 1 (HCAR1) receptor. It was recently shown that oxidative muscle fibers mainly use lactate to fuel the TCA cycle [100]. Lactate entered oxidative muscle fibers and, subsequently, the mitochondria via MCT1. Inside mitochondria lactate was oxidized to pyruvate by a mitochondrial LDH.

Others have shown that lactate is converted to pyruvate by the cytosolic LDHB [101,102]. With regard to its activity as a carbon source for oxidative metabolism, it is important to note that lactate has been reported to enhance its own internalization by upregulating the expression of the transporters MCT1 and MCT4 upon binding to cell surface receptor GPR81 [103]. Accordingly, in human U251 GBM cells under glucose deprivation [104], lactic acidosis increased lactate transport through the upregulation of MCT1 and MCT4 and converted a predominantly aerobic glycolytic into an oxidative metabolism. These changes were accompanied by the decreased expression of HIF1α and increased expression of c-MYC. Moreover, in GBM tissues, HIF1α, MCT4 and LDH were highly expressed in the interior region, while MCT1 was highly expressed in the lateral region. These results show that different tumor areas display different predominant energy-generating pathways, perhaps due to differences in vascularization and, consequently, nutrient and oxygen availability in the different areas. Exposure of cholangiocarcinoma cells for more than 2 weeks to lactic acidosis increased cell motility and enhanced respiratory capacity with an increase in mitochondrial mass [105]. Upregulation of genes associated with cell migration and EMT was also observed. The product of one of these genes, thrombospondin-1, appeared to be responsible for the increase in cell motility and respiratory capacity. Lactate (20 nM) induced metabolic reprogramming from glycolytic to oxidative metabolism in 92.1 uveal melanoma cells [106]. This was accompanied by decreased proliferation and migration, increased quiescence and euchromatin content.

In the tumor setting, lactate has been proposed to act as a carbon source for oxidative metabolism not only in an autocrine, but also in a paracrine manner, due to a cross-talk between cancer-associated fibroblasts (CAFs) and tumor cells, a phenomenon called “reverse Warburg effect” [107] (Figure 2). In this cross-talk, CAFs with a predominantly glycolytic phenotype release lactate, which is taken up by tumor cells via MCT1 and then acts as a carbon source for the TCA cycle. A clinical study investigated the expression of the lactate exporter MCT4 as a glycolysis marker and the lactate importers MCT1 and TOMM20 as markers of oxidative metabolism in 33 diffuse large B-cell lymphoma samples and in 18 non-neoplastic lymph nodes [108]. In accordance with the “reverse Warburg effect” model, TOMM20 and MCT1 were highly expressed in neoplastic lymphocytes, but not in non-neoplastic lymphocytes. Stromal cells of tumor samples strongly expressed MCT4, while stromal elements of normal lymph nodes were negative for this marker.

It should be noted, however, that another mechanism, in addition to lactate acting as a carbon source, has been proposed to underlie the “reverse Warburg effect” [109]. The addition of CAF-conditioned medium to prostate cancer cells induced posttranslational modifications of PKM2 that led to its nuclear translocation and the formation of a trimeric complex including HIF1α. This complex induced EMT and metabolic reprogramming towards oxidative metabolism in tumor cells.

Overall, the “reverse Warburg effect” concept introduces a further turn of complexity in the metabolic scenario of tumor cells. In fact, here, the metabolic heterogeneity of tumors applies to the cross-talk between the TME and tumor cells, with cells of the TME (CAFs) that instigate tumor cells to display a predominantly oxidative metabolism through the release of glycolytic metabolites, most notably lactate, which act as energetic fuels. It is still far from clear which is the overall relevance of this effect in the cancer setting, in particular, as regards the tumor types and tumor stages where it plays a significant role in shaping the metabolic make-up of tumors.

So far, we have discussed evidence showing that lactate can act as a carbon source. Lactate, however, can also upregulate oxidative metabolism by increasing mitochondrial biogenesis [110] and increasing the expression of enzymes or transporters involved in the TCA cycle and OXPHOS [104,106]. Recently, it has been shown that accumulating lactate enters the mitochondrial matrix and stimulates the ETC, leading to increased entry of pyruvate into the TCA cycle, increased OXPHOS and ATP production [111]. Increased mitochondrial ATP production downregulated glycolysis. Both L- and D-lactate enhanced ETC activity and suppressed glycolysis. Lactate appeared to stimulate the ETC by enhancing complex III electron transfer.

Lactic acidosis can also induce a variety of changes that maximize the use of substrates other than glucose. Thus, lactic acidosis was found to increase the expression of the glutamine transporter alanine, serine and cysteine transporter 2 [112,113], thereby increasing the uptake of glutamine by tumor cells. Lactic acidosis also increased glutaminolysis through the upregulation of glutaminase (GLS) 1 and GLS2. This also led to increased generation of α-KG through glutamate [114] and the replenishment of the TCA cycle. Lactic acidosis also increased FA uptake in tumor cells [115] and FAO acted as the main source of acetyl-CoA for the TCA cycle [115,116]. Another study showed that chronic exposure to acidosis of different types of cancer cells led to the upregulation of oxidative metabolism with increased FA uptake [117]. Increased FA uptake was, at least in part, responsible for the increased expression and activity of PPARα, which, in turn, upregulated the expression of genes involved in increasing mitochondrial and peroxisomal mass and β-oxidation capacity, of ETC components and negative regulators of glycolysis.

Lactic acidosis has also been reported to activate reductive glutamine metabolism in the cytosol. This was observed in tumor cells after long-term exposure to acidic pH and was supported by the upregulation of IDH1 [112]. This promoted fueling of the TCA cycle and increased mitochondrial respiration. Increased HIF2α activity upregulated the expression of transporters and enzymes involved in reductive and oxidative metabolism, while the reduced expression of HIF1α contributed to the inhibition of glycolytic metabolism. In spite of these changes, acid-adapted tumor cells had the same proliferation rate as parental cells maintained at pH 7.4. In a study performed on three breast cell lines (one normal and two cancer cell lines: MCF 10A, MCF7, MDA-MB-231) [118], it was found that high lactate levels downregulated glycolysis and induced the reductive carboxylation of glutamine to citrate in the TCA cycle. The overall fluxes remained unchanged for MCF 10A cells, but were increased in MDA-MB-231 cells. Lactic acidosis has also been shown to redirect glucose towards the oxidative branch of the PPP [114]. This effect was paralleled by the inhibition of the non-oxidative branch of the PPP and led to an overall increase in NADPH production, thereby countering the increase in ROS due to the upregulation of OXPHOS.

When talking about the effects of lactic acidosis, it should be considered that some of these effects are not due to the lactate moiety but, rather, to acidosis and, in some cases, it has been reported that both lactate as well as acidic pH are necessary in order to achieve a maximal biological effect. Thus, one consequence of lactic acidosis is the induction of resistance to glucose deprivation in tumor cells [119]. 4T1 breast cancer cells under glucose deprivation survived much longer in the presence of lactic acidosis than in its absence. Lactate alone, however, did not prolong survival, while acidosis alone was effective but appeared less potent than lactic acidosis. The same authors showed later [120] that in the presence of lactic acidosis, nine randomly selected tumor cell lines switched from a metabolism where glycolysis and oxidative metabolism contributed equally to ATP production, to a predominantly oxidative metabolism.

The observation that oxidative metabolism promotes the survival of tumor cells at a low extracellular pH is supported by results showing that the knockout of genes involved in OXPHOS and iron–sulfur cluster biogenesis underwent massive cell death in the presence of an acidic extracellular pH while growing well at a normal extracellular pH [121]. Xenografts established with OXPHOS-defective cells grew slower than control cells, but their growth was stimulated upon a reduction in the extracellular pH.

There are several other observations suggesting that some effects induced by lactic acidosis are mainly due to the acidosis rather than hyperlactemia. Thus, cancer cells cultured under conditions of lactic acidosis switched from a predominantly glycolytic to an oxidative phenotype [122] because glycolysis was severely impaired due to acidification of the cytosol. Hyperlactemia alone did not decrease the cytosolic pH and did not inhibit glycolysis. In accordance with these results, it has been observed that bicarbonate infusion into tumors attenuated tumor growth and metastases formation [123] by increasing the tumor pH and converting lactic acidosis to hyperlactemia. The results of Xie et al. [122] also suggest that both acidification of the TME as well as of the cytosol can inhibit glycolysis. In the latter case, it is unclear whether this is the consequence of a transient acidification of the cytosol, before extrusion of lactic acid out of the tumor cells, or whether there is a defect in the extrusion of lactic acid because, for example, of a lack or downregulation of the transporter that extrudes lactic acid out the cell. Inhibition of glycolysis due to acidification of the cytosol is the consequence of the inhibition of the activity of PFK1 [124] and reduced expression of glycolytic enzymes in breast cancer cells [125]. The conclusion that can be drawn from these observations is that lactic acidosis can inhibit glycolysis and upregulate oxidative metabolism by different mechanisms that are induced by either one of the two components, i.e., hyperlactemia and acidosis. This explains why in several circumstances maximal biological effects were observed in the presence of both components that build up lactic acidosis.

An intriguing role in lactic acidosis-induced upregulation of oxidative metabolism has been ascribed to the lactate receptor HCAR1 that is overexpressed in many cancer types [126]. Genetic deletion of HCAR1 in MCF7 breast cancer cells reduced the expression and activity of PFK1 and HK, and promoted a shift towards decreased glycolysis and increased oxidative metabolism. At face value, these results suggest that the expression of active HCAR1 inhibits lactate-induced upregulation of oxidative metabolism.

One consequence of the reprogramming from glycolytic to oxidative metabolism in response to lactic acidosis is the increased sensitivity of tumor cells to inhibitors of oxidative metabolism. Thus, lactic acidosis reduced glycolysis in tumor cells by 79–86% and increased OXPOS by 177–218% [127]. Accordingly, it increased the sensitivity of the cells to OXPHOS inhibitors by 2–4 orders of magnitude.

So far, we have discussed results showing that lactate and lactic acidosis upregulate oxidative metabolism. Recent evidence suggests, however, that lactate can also lead to the downregulation of oxidative metabolism. Lactylation is a posttranslational modification of Lys residues that can affect a large number of proteins [128]. Lactylation of mitochondrial proteins has been shown to inhibit OXPHOS [129]. This was the consequence of the accumulation of mitochondrial alanyl-tRNA synthetase (AARS2) occurring under hypoxic conditions. AARS2 then acted as a Lys lactyltransferase by lactylating PDH and carnitine palmitoyltransferase 2 (CPT2). This led to the inactivation of both enzymes and the inhibition of oxidative metabolism by limiting the influx of acetyl-CoA from pyruvate and FAO.

Before concluding this section on lactate and lactic acidosis, it is important to note that, recently, it has been shown that hyperlactemia in the TME may not necessarily be the result of glycolysis in tumor cells or tumor accessory cells. In fact, lactate may also be produced by intratumoral, L-lactate-producing lactic acid bacteria like *Lactobacillus iners* [130]. Its presence in cervical cancer was associated with decreased patient survival and radiation resistance. Moreover, tumor cells exposed to irradiated *L. iners* showed the upregulation of glycolysis, the TCA cycle and redox balance.

**Table 3 cells-14-01177-t003:** Molecules of glycolytic metabolism that regulate the equilibrium with oxidative metabolism.

Molecule	Consequences of Its Dysregulation (Overexpression or Inhibition)	References
HK	HK1 forms rings around mitochondria that prevented mitochondrial fission. This increases TCA activity	[80]
	O-GlcNAcylation promoted localization of HK1 on OMM, leading to increased glycolytic and mitochondrial ATP production	[81]
	Inhibition of HK2 caused glycolysis inhibition and upregulation of oxidative metabolism	[82]
	HK2 silencing in HCC cells inhibited tumorigenesis, increased cell death and oxidative metabolism, downregulated glycolysis	[83]
PGM	Persistent overexpression of PGM led to increased levels of glycolytic metabolites and inhibition of mitochondrial respiration. Lactate levels remained unchanged	[86,87]
PKM	Active PKM2 increased oxalacetate levels, which inhibited LDHA in tumor cells and caused reduced lactate production	[90]
	Knockdown of PKM1 and PKM2 in H1299 lung cancer cells stimulated mitochondrial biogenesis and preserved ATP levels. This did not occur in A549 cells and ATP levels declined	[91]
	In TNBC cells, PKM2 inhibited FAO. Inhibition of PKM2 induced reprogramming from glycolysis towards FAO	[92]
Lactic acidosis	Lactate is a carbon source for the TCA cycle	[96,97,98]
	Lactate entered the mitochondrial matrix and stimulated the ETC with increased ATP production	[111]
	Lactate increased glutaminolysis with increased αKG production	[112,113,114]
	Lactic acidosis also increased FA uptake, FAO and acetyl-CoA generation for the TCA cycle	[114,115,116]
	High levels of lactate inhibited glycolysis	[100]
	Lactate upregulated expression of lactate transporters MCT1 and MCT4	[103,104]
	Lactic acidosis increased motility and enhanced respiratory capacity of cholangiocarcinoma cells	[105]
	Lactate induced switch from glycolysis to oxidative metabolism in melanoma cells	[106]
	Lactic acidosis induced switch from glycolysis to oxidative metabolism due to acidification of the cytosol	[122]
	Acidification of cytosol inhibited glycolysis through inhibition of PFK1 and reduced expression of glycolytic enzymes	[123,124]
	Exposure to acidosis of different cancer cells upregulated oxidative metabolism with increased FA uptake	[117]
	Lactic acidosis activated reductive glutamine metabolism in the cytosol	[112,118]
	Lactic acidosis induced resistance to glucose deprivation in tumor cells	[119]
	Lactylation of mitochondrial proteins inhibited OXPHOS	[129]

Abbreviations: αKG, α-ketoglutarate; ATP, adenosine triphosphate; ETC, electron transport chain; FA, fatty acid; FAO, FA oxidation; HCC, hepatocellular carcinoma; HK, hexokinase; LDHA, lactate dehydrogenase; MCT, monocarboxylate transporter; OMM, outer mitochondrial membrane; OXPHOS, oxidative phosphorylation; PFK, phosphofructokinase; PGM, phosphoglycerate mutase; PKM, pyruvate kinase M; TCA, tricarboxylic acid; TNBC triple negative breast cancer.

## 6. Manipulating the Balance Between Oxidative and Glycolytic Metabolism by Pharmacologic or Genetic Means

In this section we discuss some representative examples of the consequences of inhibiting the activity or downregulating the expression of components of the two energy-generating pathways of cellular metabolism.

### 6.1. Inhibitors of Oxidative Metabolism

Inhibition of oxidative metabolism by pharmacological or other interventions has been shown to upregulate glycolysis.

Tigecycline is an anti-bacterial agent that inhibits mitochondrial protein synthesis, causing cell death. A leukemic cell line resistant to tigecycline was established and investigated [131]. The resistant cells showed upregulated glycolysis, reduced oxygen consumption, undetectable levels of mitochondrially translated proteins and abnormally swollen mitochondria. Upon the removal of tigecycline, the cells reestablished oxidative metabolism, but returned to a predominantly glycolytic metabolism upon retreatment with tigecyline for 72 h. This is a striking example of the metabolic plasticity of a tumor cell line.

Another study showed that H460 and A549 tumor cells upregulated glycolysis upon the inhibition of oxidative metabolism with compounds like rotenone and 2-dinitrophenol [132]. On the other hand, these cell lines did not upregulate oxidative metabolism upon the inhibition of glycolysis, suggesting that, in these cases, glycolysis was more flexible and adaptable than oxidative metabolism. Interestingly, the upregulation of glycolysis was mediated by AMPK, showing that, at least in this setting, glycolysis upregulation was not a compensatory mechanism coming into play upon the inhibition of oxidative metabolism, but was actively induced upon the activation of AMPK.

LNT-229 GBM cells without functional mitochondria and suppressed oxidative metabolism showed a glycolytic phenotype that promoted resistance towards anti-vascular endothelial growth factor (bevacizumab) therapy in experimental tumors [133].

Biguanides are probably the best-known class of inhibitors of oxidative metabolism. The biguanide metformin (1,1-dimethylbiguanide hydrochloride) is the most widely used first-line drug for the treatment of type II diabetes. While numerous mechanisms of action have been ascribed to metformin, the most broadly accepted is its inhibitory action on the ETC complex I of mitochondrial respiration [134]. This effect has several consequences that encompass both antiglycolytic as well as proglycolytic effects [135]. In fact, the inhibition of OXPHOS causes a decrease in the ATP/AMP ratio. Increased AMP levels induce the activation of AMPK [136]. Activated AMPK inhibits mTOR [137]. Since mTOR activates several transcription factors that upregulate the expression of glucose transporters and glycolytic enzymes [138], the mTOR-inhibitory activity of AMPK can downregulate glycolysis. On the other hand, AMPK has been shown to upregulate glycolysis upon the phosphorylation and activation of PFK2, which synthesizes fructose 2,6-biphosphate, a glycolysis inducer [139]. In addition, the inhibitory effect of metformin on oxidative metabolism can upregulate glycolysis through a compensatory mechanism, as described in other parts of this manuscript. It is unclear which conditions promote the predominance—the proglycolytic or the antiglycolytic effect. Perhaps, the concurrent activation of HIF-1α under hypoxic or pseudohypoxic conditions may override the inhibitory effect of AMPK on mTOR and/or increasing doses of metformin may shift the balance towards glycolysis. In fact, the most dangerous and even life-threatening side effect of metformin, lactic acidosis, which is the consequence of an upregulated, systemic glycolysis, is observed at increasing doses of metformin [140].

An antitumor activity of metformin has been suggested by the observation that its use was associated with a lower risk of cancer incidence in diabetic patients [141]. These observations have led to the launch of a large number of clinical trials with metformin or the related biguanide phenformin in cancer patients. The results published so far are not encouraging [142] but the results of further studies are awaited.

Other molecules with inhibitory effects on oxidative metabolism have also been shown to upregulate glycolysis. Thus, a lipophilic cation, VLX600, reduced mitochondrial respiration [143]. It was cytotoxic to CRC cells, particularly under conditions of nutrient starvation, and was preferentially active on quiescent cells of tumor cell spheroids. VLX600 induced a metabolic shift towards glycolysis that was dependent of HIF1α. α. VLX600 has been tested in early-phase clinical trials [144].

Mitochondrial uncouplers, either exogenous or endogenous [145], can also upregulate glycolysis. As regards endogenous uncouplers, mesenchymal stromal cells have been shown to upregulate the expression of a member of the uncoupling family of proteins (uncoupling protein 2) in leukemia cells [146]. Mitochondrial uncoupling by this protein led to increased aerobic glycolysis.

### 6.2. Inhibitors of Glycolysis

Dichloroacetate (DCA) induces the downregulation of glycolysis and a shift to oxidative metabolism through different mechanisms of action, the most prominent being the inhibition of PDK, which leads to the upregulation of PDH activity and, consequently, to increased activity of the TCA cycle and oxidative metabolism [147]. DCA added to cisplatin-resistant human head and neck cancer (HNC) cells caused PDK2 overexpression and downregulated glycolysis [148]. It induced HNC cell death by decreasing the mitochondrial membrane potential and enhancing mitochondrial ROS production. Moreover, the induction of glucose oxidation activated mitochondrial apoptotic signaling and cell death.

Some early-phase clinical trials have been performed with DCA, as well as with other inhibitors of glycolytic enzymes (e.g., the HK inhibitor 2DG), but these products have not advanced further because of an unfavorable side effect profile and/or lack of efficacy. The only candidate drug that is still in active development is the MCT1 inhibitor AZ3965 [149].

An important article on the balance between glycolytic and oxidative metabolism and the consequences of disrupting this balance has been published most recently [150]. In this work, Chinese hamster ovary (CHO) cells were generated, in which the genes encoding for LDHs as well PDKs had been knocked out. These cells were no longer able to generate lactate from pyruvate and pyruvate was converted to acetyl-CoA and entered the TCA cycle in the absence of any negative control of PDKs on PDH. The resulting CHO cells remained rapidly proliferative in spite of showing decreased glycolytic flux. On the other hand, the cells channeled more glucose into oxidative metabolism as witnessed by the increased generation of TCA cycle metabolites. Thus, lactate production was not required by these cells to meet energetic or biosynthetic demands. This does not exclude that crucial glycolytic intermediates for biomass production or ATP were generated before pyruvate conversion or that differences in terms of survival or proliferation may have become apparent under particular environmental conditions, such as hypoxic or pseudohypoxic conditions or under conditions promoting accelerated proliferation [151].

## 7. Conclusions

In this article we have reviewed the mechanisms underlying the reciprocal control between the two energy-producing pathways of the cell, i.e., oxidative metabolism and glycolysis. A first conclusion that can be drawn is that tumors are able to exploit or circumvent these control mechanisms at their own profit. Thus, metabolic reprogramming, whether from glycolysis to oxidative metabolism or vice versa, can facilitate tumor initiation and progression [152]. These phenotypic changes add to other changes, like EMT and autophagy [153,154,155,156], that tumor cells can adopt in order to survive, proliferate and metastasize in response to environmental stressors. Metabolic reprogramming also operates in mediating resistance to a large number of antitumor therapies [19].

Another important aspect to consider is the large number of mechanisms that have been put in place in order to maintain the equilibrium between oxidative and glycolytic metabolism and, consequently, the equally large number of mechanisms that can become dysregulated in pathological situations, most notably cancer.

An intriguing issue that emerges from the mechanisms that have been described is that, in some cases, one energy-producing pathway becomes, at least apparently, upregulated in a compensatory manner, as a consequence of the downregulation/inhibition of the other pathway (e.g., Refs. [31,47]). In other cases, however, the downregulation/inhibition of the other pathway appears to be the result of an active effect, in some cases mediated by the same factor that leads to the upregulation of the first pathway (e.g., Refs. [27,36]). These observations raise several questions: are compensatory mechanisms really operative or have active mechanisms simply been overlooked? Vice versa, why do active mechanisms actually exist if compensatory mechanisms can achieve the same goal? These are important questions in order to further elucidate the mechanisms that control the equilibrium between glycolytic and oxidative metabolism.

As can be seen from this article, the overproduction of lactate and the development of lactic acidosis, and the biological effects that derive therefrom, have taken center stage in tumor metabolism and in the equilibrium between oxidative and glycolytic metabolism. As herein discussed, lactate acts as an important carbon source for oxidative metabolism and it can also upregulate oxidative metabolism through other mechanisms that have been discussed. The most interesting conclusion that can be drawn from these observations is that tumor metabolism steers towards heterogeneity and when one pathway becomes predominant over the other, tumor metabolism tries to reinstate a more balanced equilibrium, which is likely more profitable for tumor growth and dissemination.

The findings on the role of lactate and lactic acidosis are relatively recent and have accumulated in the last few years. It is therefore not surprising that there are still many open questions. Thus, in many cases, it is not yet entirely clear which effects are mediated by intracellular or extracellular hyperlactemia and acidosis, as well as which effects are mediated upon lactate import and which are mediated by cell surface receptors, most notably GRP81/HCAR1.

From a therapeutic point of view, blocking metabolic reprogramming by acting on the control mechanisms that we have described is certainly a promising avenue that is worthwhile exploring for cancer therapy. This might yield additive or even synergistic effects when combined with currently available antitumor therapies. In view of the multitude of control mechanisms that have been identified so far, however, it appears necessary to have available suitable biomarkers, allowing identifying the exact mechanism(s) that is or are operative at any given time with the caveat that different mechanisms may be at work at different times during tumor progression. While this is a road that needs to be explored not only for the identification of new antitumor therapies but also to expand our knowledge about the basic, metabolic mechanisms that support tumorigenesis, it appears, at present, more promising to act on both energy-producing pathways in order to block tumor-associated metabolic derangements and reprogramming. Moreover, as discussed in the initial part of this article, tumor cells are in a hybrid metabolic state in which both glycolytic as well as oxidative metabolism coexist for energy production and the generation of intermediates for biomass production. The need to act on both metabolic pathways has already been demonstrated, for example, with TNBC cells [157]. Such an approach, however, may be burdened by intolerable side effects, in particular, when the corresponding therapeutic(s) has or have to be administered over prolonged periods of time or even chronically, as may be the case for tumor therapy. For this purpose, the possibility of tumor-specific delivery should be considered. Such an approach has been very successful in the delivery of antitumor therapeutics, mainly through the use of monoclonal antibodies recognizing tumor-associated or tumor-specific molecular targets ([158,159,160,161]). Similar approaches have been investigated only marginally for the specific delivery of metabolic inhibitors (e.g., [162,163,164]). It appears timely to explore them more deeply in for cancer therapy.

## Figures and Tables

**Figure 1 cells-14-01177-f001:**
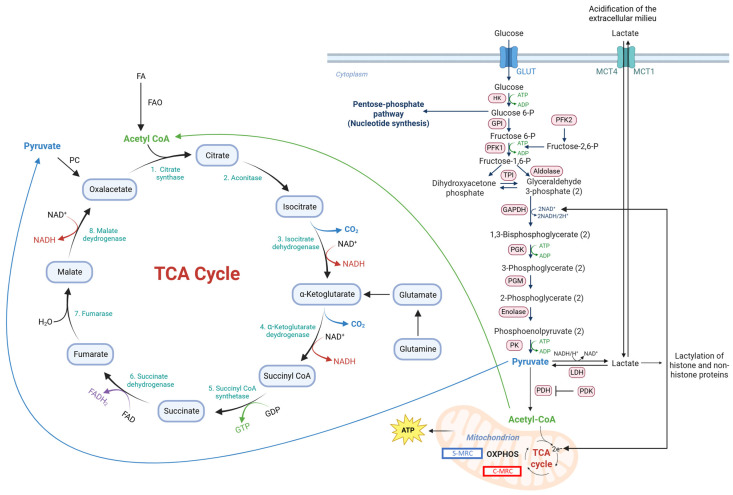
Representation of oxidative and glycolytic metabolism and some mechanisms that regulated the equilibrium between the two. Abbreviations: ATP, adenosine triphosphate; FA, fatty acid; FAD, flavin adenine dinucleotide, oxidized; FADH2, flavin adenine dinucleotide, reduced; FAO, FA oxidation; GAPDH, glyceraldehyde 3-phosphate dehydrogenase; GDP, guanosine diphosphate; GTP, guanosine triphosphate; GLUT, glucose transporter; GPI, glucose 6-phosphate isomerase; HK, hexokinase; LDH, lactate dehydrogenase; MCT, monocarboxylate transporter; MRC, mitochondrial respiratory complex; NAD^+^, nicotinamide adenine dinucleotide, oxidized; NADH, nicotinamide adenine dinucleotide, reduced; OXPHOS, oxidative phosphorylation; PC, pyruvate carboxylase; PDH, pyruvate dehydrogenase; PDK, pyruvate dehydrogenase kinase; PFK, phosphofructokinase; PGK, phosphoglycerate kinase; PGM, phosphoglycerate mutase; PK, pyruvate kinase; ROS, reactive oxygen species; TCA, tricarboxylic acid; TPI, triosephosphate isomerase.

**Figure 2 cells-14-01177-f002:**
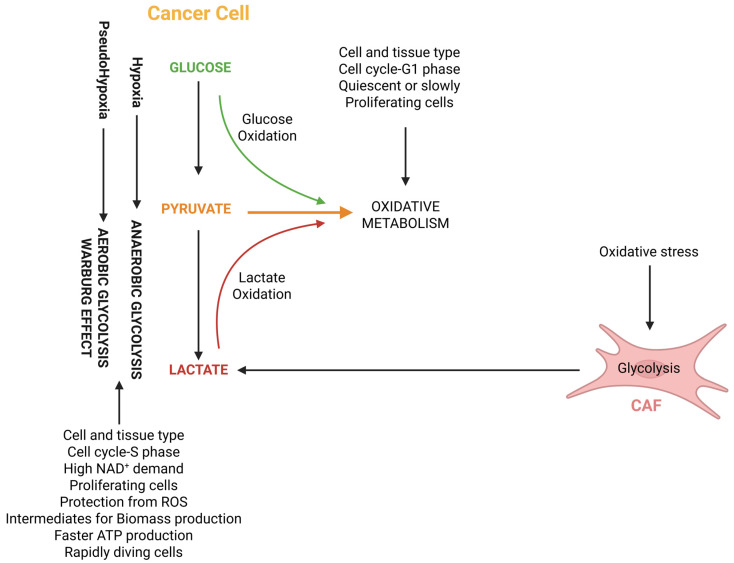
The relationship between anaerobic and aerobic (Warburg effect) glycolysis, reverse Warburg effect and oxidative metabolism. The figure also shows some of the features that promote a predominantly oxidative or glycolytic metabolism. Abbreviations: CAF, cancer-associated fibroblast; NAD^+^, nicotinamide adenine dinucleotide, oxidized; ROS, reactive oxygen species.

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
