# Peer review of "Oxidative and Glycolytic Metabolism: Their Reciprocal Regulation and Dysregulation in Cancer"

_cells, 2025, doi:10.3390/cells14151177_

Round 1
Reviewer 1 Report
Comments and Suggestions for Authors
The manuscript by Cordani et al. provides a comprehensive review of the intricate regulation between oxidative and glycolytic metabolism in the context of cancer. The authors highlight metabolic heterogeneity, plasticity, and the emerging role of lactic acidosis as a central regulator of energy metabolism in various cancerous conditions. The references summarized by the authors challenge the oversimplified narrative of the Warburg effect by demonstrating that oxidative metabolism remains important in many tumors.
The strengths of the manuscript include:
- A comprehensive scope of the coverage of relevant elements pertaining to cancer metabolism and the integration of classical metabolic pathways.
- A detailed discussion of key molecular nodes and their dysregulation in cancer;
- Addressing controversies with balanced critiques.
- Clear summary tables of key molecules within relevant sections;
- Inclusion of potential therapeutic implications.
Despite these merits, the manuscript could be improved in the following aspects:
- Keywords are missing.
- While the introduction provides a solid overview of classical energy metabolism, it could be better organized to explicitly connect with the individual sections of the review.
- The conceptual models could be refined. A schematic summarizing the Warburg Effect, the Reverse Warburg Effect, and hybrid metabolic models in the context of cancer would enhance clarity.
- The section on IDH mutations could be better structured to reconcile conflicting findings in the literature.
- The discussion of the Reverse Warburg Effect lacks critical depth, despite ongoing debate in the field.
Overall, this review provides a comprehensive update on Warburg-centric perspectives of cancer metabolism, which could serve as a valuable reference for researchers in cancer biology and metabolic studies. Addressing the above concerns will strengthen the manuscript.
Author Response
Answers to the points raised by Reviewer 1.
Criticism: Keywords are missing.
Answer: Keywords have now been included. They are on the title page, just after the Abstract.
Criticism: While the introduction provides a solid overview of classical energy metabolism, it could be better organized to explicitly connect with the individual sections of the review.
Answer: We have now modified the Abstract and the Introduction in order to establish a closer link to the sections of the manuscript.
Criticism: The conceptual models could be refined. A schematic summarizing the Warburg Effect, the Reverse Warburg Effect, and hybrid metabolic models in the context of cancer would enhance clarity.
Answer: We have now included a new figure (Figure 2) to address this point.
Criticism: The section on IDH mutations could be better structured to reconcile conflicting findings in the literature.
Answer: We have almost entirely rewritten this section. We have put together references showing glycolysis-inhibitory effects and, separately, articles showing glycolysis-stimulatory effects. In order the address a similar point raised by reviewer 2 we have also included a paragraph with a tentative explanation of these contradictory effects (see lines 442-474).
Reviewer 2 Report
Comments and Suggestions for Authors
I have reviewed the article titled "Oxidative and Glycolytic Metabolism: Their Reciprocal Regulation and Dysregulation in Cancer”. I have the following concerns:
- The review is largely descriptive and less critical.
- There should be more discussion on conflicting findings, experimental limitations, and gaps in current knowledge.
- For example: IDH mutations are presented with conflicting outcomes—this could be critically compared rather than simply reported.
- While the manuscript is a review, it sometimes relies on citation chains without specifying pivotal primary studies.
- Suggest incorporating more landmark primary studies directly to strengthen the narrative.
- The manuscript could significantly benefit from graphical abstracts, summary figures, and pathway diagrams.
- Current figures are sparse and heavily text-based. Inclusion of schematics showing the metabolic interplay would improve clarity and engagement.
- The therapeutic implications are briefly mentioned but not comprehensively discussed.
- The authors could integrate more about targeted therapies, ongoing clinical trials, and potential metabolic vulnerabilities.
- Suggest adding a dedicated section on clinical translation and future research directions.
- Some minor grammatical errors and typographical inconsistencies (e.g., missing article in "lactosis" which should probably be "lactate accumulation" or "hyperlactatemia").
- Some figure and table captions are incomplete or could be more explanatory.
Due to these gaps, I recommend Major Revision before this manuscript can be considered for publication.
Comments on the Quality of English LanguageSome minor grammatical errors and typographical inconsistencies
Author Response
Criticism: The review is largely descriptive and less critical.
Answer: The reviewer is right. It must be considered, however, that, to the best of our knowledge, this is the first publication that addresses the mechanisms that regulated the equilibrium between glycolytic and oxidative metabolism and their dysregulation in cancer.
Criticism: There should be more discussion on conflicting findings, experimental limitations, and gaps in current knowledge. For example: IDH mutations are presented with conflicting outcomes—this could be critically compared rather than simply reported.
Answer: We have now added several new paragraphs in order to address this point: one on IDH mutations (lines 442-474), one in the discussion on compensatory vs active modifications of the equilibrium between the energy-producing pathways (lines 1041-1056).
Criticism: While the manuscript is a review, it sometimes relies on citation chains without specifying pivotal primary studies. Suggest incorporating more landmark primary studies directly to strengthen the narrative.
Answer: We presume that the reviewer refers mainly to the final section. In order to address this point, we have now included references of several landmark studies (refs. 156, 159-161).
Criticism(s): The manuscript could significantly benefit from graphical abstracts, summary figures, and pathway diagrams. Current figures are sparse and heavily text-based. Inclusion of schematics showing the metabolic interplay would improve clarity and engagement.
Answer: As already indicated in the reply to Reviewer 1, in order to address these points, we have now added a new figure (Figure 2) that illustrates the relationship of glycolytic and oxidative metabolism, the stimuli that induce the predominance of one over the other, and the relationship of the two with the reverse Warburg effect.
Criticism(s): The therapeutic implications are briefly mentioned but not comprehensively discussed. The authors could integrate more about targeted therapies, ongoing clinical trials, and potential metabolic vulnerabilities. Suggest adding a dedicated section on clinical translation and future research directions.
Answer: We have now expanded the sections discussing inhibitors of glycolytic or oxidative metabolism and the results that have been obtained so far in clinical trials (lines 971-975, 1009-1012 with some new references). We would like to point out, however, that we included the pharmacological issues mainly because they are an excellent example how a compensatory upregulation of a pathway is induced upon inhibition of the other pathway. Our goal was not to review the current development status of inhibitors of glycolytic or oxidative metabolism. There are excellent, recent reviews that make point on this issue
Criticism(s): Some minor grammatical errors and typographical inconsistencies (e.g., missing article in "lactosis" which should probably be "lactate accumulation" or "hyperlactatemia"). Some figure and table captions are incomplete or could be more explanatory.
Answer: We have accurately gone through the manuscript and amended typos.
Reviewer 3 Report
Comments and Suggestions for Authors
This manuscript is a review that introduces research on the metabolic switching between glycolysis and oxidative phosphorylation for ATP production in cancer. Although their regulatory mechanisms are highly complex due to their diversity, the review provides a well-organized summary. In particular, the explanation of the switching between the two pathways mediated by lactate is well structured. In my opinion, the manuscript should be accepted after minor revision.
#Comments
- The importance of enhanced activity of HK and PFK in promoting glycolysis is mentioned several times, but there is little explanation on the regulation of PFK. Although the authors state that textbook-level feedback mechanisms are not the scope of the review, PFK is a key enzyme in the regulation of glycolysis. Therefore, like HK, a dedicated section should be included to explain PFK, including brief descriptions of feedback mechanisms, expression, and activity regulation by focusing on the specific regulation in cancer cells.
- Line 254: It is unclear why a general DNA relaxation enzyme (topoisomerase I) is related to PDK expression. It is better to add a brief explanation of whether the mechanism if it has been already shown or some interpretation.
- In the latter half of the manuscript, typos were found. There may be more than those listed below, so a thorough check is recommended.
Line 630: "slower that" → "slower than"?
Line 710: "have ben" → "have been"?
Double spaces? e.g., Line 585: “ Lactate…” -
Fig. 1.: The arrow for NAD+ NADH conversion in the Malate dehydrogenase reaction is opposite.
- Abbreviations are listed in each table, but it might be better to consolidate them in the last abbreviation section of the manuscript. Also, since there is sufficient space, it would be more reader-friendly to indicate, like “Hexokinase (HK)” in the “Molecule or molecular complex” column.
Author Response
Criticism: The importance of enhanced activity of HK and PFK in promoting glycolysis is mentioned several times, but there is little explanation on the regulation of PFK. Although the authors state that textbook-level feedback mechanisms are not the scope of the review, PFK is a key enzyme in the regulation of glycolysis. Therefore, like HK, a dedicated section should be included to explain PFK, including brief descriptions of feedback mechanisms, expression, and activity regulation by focusing on the specific regulation in cancer cells.
Answer: We have now included a new section on PFK1 (section 5.2 of the present version).
Criticism: Line 254: It is unclear why a general DNA relaxation enzyme (topoisomerase I) is related to PDK expression. It is better to add a brief explanation of whether the mechanism if it has been already shown or some interpretation.
Answer: The reviewer is absolutely right. Unfortunately, the original paper did not give an explanation. We have now included a tentative explanation of ours.
Criticism: In the latter half of the manuscript, typos were found. There may be more than those listed below, so a thorough check is recommended.
Line 630: "slower that" → "slower than"?
Line 710: "have ben" → "have been"?
Double spaces? e.g., Line 585: “ Lactate…”
Answer: We have accurately gone through the manuscript and amended typos.
Criticism: Fig. 1.: The arrow for NAD+ NADH conversion in the Malate dehydrogenase reaction is opposite.
Answer: We have checked, but the arrow should be OK. The reaction is:
L- Malate + NAD+ ® oxalacetate + NADH + H+ (we have omitted the protons from the figure).
Criticism: Abbreviations are listed in each table, but it might be better to consolidate them in the last abbreviation section of the manuscript. Also, since there is sufficient space, it would be more reader-friendly to indicate, like “Hexokinase (HK)” in the “Molecule or molecular complex” column.
Answer: There is, in fact, an abbreviation section. It is at the end of the text, before the references.
Round 2
Reviewer 2 Report
Comments and Suggestions for Authors
The authors have carefully addressed and incorporated all the suggestions provided by the reviewers. The necessary revisions have been made to enhance the quality and clarity of the manuscript. After a thorough evaluation, I find the revised version satisfactory and suitable for publication.
Therefore, I recommend the manuscript for publication in its current form.